# Jelleine, a Family of Peptides Isolated from the Royal Jelly of the Honey Bees (*Apis mellifera*), as a Promising Prototype for New Medicines: A Narrative Review

**DOI:** 10.3390/toxins16010024

**Published:** 2024-01-02

**Authors:** William Gustavo Lima, Julio Cesar Moreira Brito, Rodrigo Moreira Verly, Maria Elena de Lima

**Affiliations:** 1Programa de Pós-Graduação Stricto Sensu em Medicina e Biomedicina, Faculdade de Saúde da Santa Casa de Belo Horizonte, Avenida dos Andradas, 2688, Santa Efigênia, Belo Horizonte 30110-005, MG, Brazil; lima.mariaelena@gmail.com; 2Fundação Ezequiel Dias (FUNED), Rua Conde Pereira Carneiro, 8, Gameleira, Belo Horizonte 30510-010, MG, Brazil; julio.brito@funed.mg.gov.br; 3Departamento de Química, Faculdade de Ciências Exatas, Universidade Federal dos Vales do Jequitinhonha e Mucuri, Rodovia MGT 367, 5000, Auto da Jacuba, Diamantina 39100-000, MG, Brazil; verly.rodrigo@gmail.com

**Keywords:** jelleine-I, royal jelly, antibacterial, antifungal, anti-leishmania, anti-inflammatory, healing

## Abstract

The jelleine family is a group of four peptides (jelleines I–IV) originally isolated from the royal jelly of honey bee (*Apis mellifera*), but later detected in some honey samples. These oligopeptides are composed of 8–9 amino acid residues, positively charged (+2 to +3 at pH 7.2), including 38–50% of hydrophobic residues and a carboxamide *C*-terminus. Jelleines, generated by processing of the *C*-terminal region of major royal jelly proteins 1 (MRJP-1), play an important biological role in royal jelly conservation as well as in protecting bee larvae from potential pathogens. Therefore, these molecules present numerous benefits for human health, including therapeutic purposes as shown in preclinical studies. In this review, we aimed to evaluate the biological effects of jelleines in addition to characterising their toxicities and stabilities. Jelleines I–III have promising antimicrobial activity and low toxicity (LD_50_ > 1000 mg/Kg). However, jelleine-IV has not shown relevant biological potential. Jelleine-I, but not the other analogues, also has antiparasitic, healing, and pro-coagulant activities in addition to indirectly modulating tumor cell growth and controlling the inflammatory process. Although it is sensitive to hydrolysis by proteases, the addition of halogens increases the chemical stability of these molecules. Thus, these results suggest that jelleines, especially jelleine-I, are a potential target for the development of new, effective and safe therapeutic molecules for clinical use.

## 1. Introduction

Bee products such as honey, propolis, royal jelly, bee pollen, beeswax and bee venom contain a large number of bioactive components; the use of these substances for disease prevention or treatment is called apitherapy [1]. In the past, apitherapy products were frequently used as natural remedies and their roots can be traced back to ancient Egypt more than 6000 years ago [2]. Recently, bee products have been newly incorporated into medical practice with a focus on isolating bioactive molecules with various pharmacological properties, especially antimicrobial [3,4,5], anti-inflammatory [6], healing [7,8], antioxidant/antiaging [9], and antitumoral properties [10]. 

Among apitherapy products, royal jelly (RJ) stands out as one of the most important sources of bioactive compounds, including proteins, peptides, phenolic compounds, flavonoids, heterocyclic compounds, lipids and complex carbohydrates, which have high biological potential [8]. RJ is the yellowish-white and creamy substance secreted in the hypopharyngeal and mandibular glands of worker bees. This mixture is used as food for all honey bee larvae in the first three days after birth. After three days, worker bee larvae start eating worker bee jelly (also called bee bread), which is a mixture of honey, pollen and nectar, and queen bee larvae continue to eat RJ [11]. Due to the biological importance of RJ as a nutritional factor for bee societies, this substance has numerous therapeutic properties for humans as a raw material widely used as a functional food and food supplement [12].

RJ is susceptible to colonisation by various microorganisms (bacteria and yeasts) often associated with hive products such as honey, pollen and propolis [13]. Since the contamination of RJ endangers larval health and affects food quality, bees have developed different methods of preserving this product such as through the production of molecules with antimicrobial properties [14]. Although insects do not have a lymphocyte-based immune system, they can successfully eliminate microorganisms, suggesting that these invertebrates produce very effective antimicrobial substances [15,16]. In fact, some proteins and peptides secreted by bees have high antimicrobial activity and are added to products such as RJ in order to preserve this substance and help in the innate immune response of larvae against infections [14,16,17].

Besides the nutritional functions, long-peptide royalisin (40 amino acids) and the major royal jelly proteins (MRJP-1-9) isolated from RJ have presented important antimicrobial properties [14]. In addition, a group of four short peptides named jelleines (jelleine I–IV) have been identified in RJ. The antimicrobial activity of jelleines I–III has been described [13,18,19,20,21,22,23,24,25,26,27,28], but jelleine IV shows no antibacterial or antifungal activity [13]. These peptides consist of 8–9 amino acid residues, which are 38–50% hydrophobic residues, containing a carboxyamide *C*-terminus and which carry a positive charge (+1 or +2) at physiological pH (7.2) [13]. Recently, some studies have shown that jelleines present additional pharmacological activities, such as anti-inflammatory [21,29], antiparasitic [30], healing [31], haemostatic [32] and antitumor [22]. In this review, we further describe the potential pharmacological uses of jelleines. In addition, we provide an overview of the main analogues as well as the main technical and chemical strategies used to optimise the action of this peptide family, highlighting the modifications aimed at maximising biological activity while minimising toxic effects. Thereafter, we discuss the stability of jelleines and the tools used to improve them to enable their therapeutic use in vivo. 

## 2. History and Origin of Jelleines

Jelleines were first isolated in 2004 by the research group coordinated by Professor Mario Sergio Palma at the Universidade Estadual de São Paulo (UNESP), Rio Claro, São Paulo, Brazil [13]. Fontana et al. [13] collected samples of RJ from 3-day-old larvae of Africanised honey bees (*Apis mellifera*) and purified the mixtures by reversed-phase high-performance liquid chromatograph (RP-HPLC). A total of 10 fractions were manually collected and biologically monitored through the evaluation of the antibacterial activities. Only two fractions (named 6 and 8) showed antimicrobial activity against gram-positive and gram-negative bacteria, which were subsequently characterised via mass spectrometry. Interestingly, the presence of four different peptides with molecular weights ranging from 942 to 1082 Da was detected. This mixture was fractionated in a second step providing an RP-HPLC profile with four solved peaks, which were collected and identified by mass spectrometry (Q-TOF-MS/MS). Finally, the authors reported four unpublished peptides containing the carboxamide *C*-terminal, named jelleines (I–IV) [13]. The name, sequence, and some physicochemical proprieties of these peptides are listed in Table 1.

Amino acid sequence analysis showed no similarity of the jelleines to other antimicrobial peptides (AMPs) produced by honey bees, such as hymenoptaecin, apidaecin, abaecin, melittin, apamin, and royalisin [13]. However, the primary sequences of the jelleines showed high similarity with the last nine amino acid residues of the *C*-terminal region of MRJP-1, a common protein present in the RJ. The proteins account for 17 to 45% of the dry matter of RJ, with more than 80% of the protein content being accounted for by the MRJPs group [12]. According to Drapeau et al. [33], the genome of *A. mellifera* encodes nine MRJPs, namely MRJP-1 to MRJP-9. These proteins are primarily associated with the specific physiological action of RJ in queen bee development, as they provide the essential amino acids for larval development [34]. MRJP-1 is a weakly acidic glycoprotein that accounts for 31–66% of total RJ proteins and its architecture includes both monomeric (55 kDa; also called royalactin) and oligomeric (280–420 kDa; also called apalbumin) forms [35]. Besides its major importance in honey bees, MRJP-1 has a variety of pharmaceutical effects on human health, such as antioxidant [36], healing and cell proliferating [37], antibacterial [38], antihypertensive [39], immunomodulatory and anti-inflammatory activities [40], hypocholesterolemic [41], neuroprotective [42] and reproductive and fertility-promoting activities [43].

According to Fontana et al. [13], the presence of the arginine residue at position 373 adjacent to the threonine residue (position 374) in the primary sequence of MRJP-1, suggests that jelleine-II is derived from the hydrolysis of MRJP-1 by trypsin present in RJ. Subsequently, unknown exoproteinases act on the *N*-terminal and *C*-terminal positions of the jelleine-II, resulting in the formation of the jelleine-I and jelleine-IV, respectively [13]. The origin of jelleine-III, in turn, is currently unclear [13] (Figure 1 and Figure 2). 

In 2015, Brudzynski and Sjaarda [44] suggested that jelleines may be present in other bee products besides RJ. The authors identified MRJP-1 in purified glycoprotein concentrates (G) from honey (H). In this study, concavalin A chromatography was used to isolate glycoproteins from four honey samples (H177, H207, H208, H210) donated by Canadian beekeepers. Sequencing analysis revealed three samples (H177, H208 and H210) containing MJRP-1 in their composition. All glycoprotein concentrates (but especially G208 and G210) showed antibacterial activity against *Escherichia coli* and *Bacillus subtilis*, suggesting two well-defined effects for this mannose-rich mixture: (i) a selective effect on bacterial cells, leading to their agglutination; and (ii) less specific membrane permeabilisation, acting in both bacterial cells and erythrocytes [44]. However, deglycosylation of two glycoprotein concentrates (G207 and G208) had a drastically different effect on their antibacterial activities. Therefore, the deglycosylation of G207 resulted in a significant loss of antibacterial activity, whereas deglycosylated G208 retained its biological activity. MALDI-TOF sequencing of the major protein in G207 revealed the prevalence of MRJP-2 protein in this mixture. Unlike MRJP-1, MRJP-2 does not contain encrypted AMPs in its sequence, suggesting jelleines as the main antibacterial agents of honey containing MRJP-1 (i.e., H177, H208 and H210) [44]. Later, the presence of jelleines in honey was confirmed by Leiva-Sabadini et al. [45], revealing the presence of jelleine-III in exosome-like extracellular vesicles isolated from single-flowered honey of *Eucryphia cordifolia* Cav. (Cunoniaceae) obtained from Chilean beekeepers.

**Figure 2 toxins-16-00024-f002:**
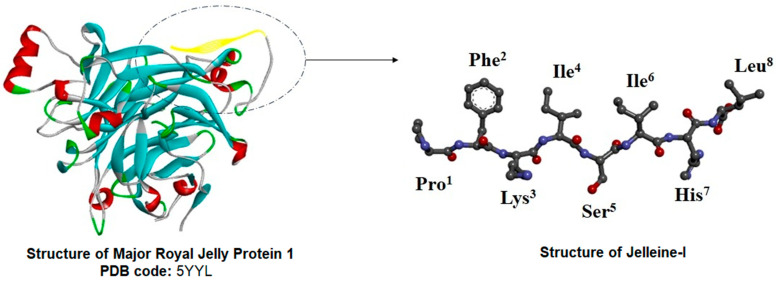
Tertiary structure of Major Royal Jelly Proteins (MRJP)-1 (PDB: 5YYL; [46]), highlighting the jelleine-I sequence (in yellow).

## 3. Chemical Properties of Jelleines 

All jelleines have a sequence of seven conserved amino acid residues (i.e., PFKISIH) (Table 1), which differ by modifications in the *C*-terminal or *N*-terminal region of this sequence. Jelleines I–III have a leucine residue added to the *C*-terminal portion, and all exhibit some degree of antimicrobial activity in vitro [13,18]. These data suggest that leucine is an important pharmacophore group in this peptide family. This hypothesis is supported by the fact that the removal of the leucine residue, as observed in the jelleine-IV sequence, is sufficient to completely suppress the antimicrobial activity of this compound [13]. Indeed, the presence of leucine in the amidated form at the *C-*terminal position of the cationic oligopeptides obtained from the toxins of insects of the order Hymenoptera [47,48,49] and of most proline-rich peptides from insects in general [16] appears to be a mandatory structural requirement for the antimicrobial activity of these natural products.

Jelleine-I and II present similar antimicrobial activities [13,18] and circular dichroism analyses show similar conformational behavior for both peptides. In an aqueous solution (phosphate buffer, pH 7), jelleine-I and II are in a random coil conformation, acquiring a helical conformation only in the presence of mimetic media of anionic membranes (sodium dodecyl sulphate; SDS) [18]. Therefore, the presence of a threonine residue in the *N*-terminal region does not seem to be essential for the biological activity of jelleines. On the other hand, the presence of glutamate in the *N*-terminal region of jelleine-III significantly reduces both solubilities in phosphate buffer and antimicrobial activity. In addition, jelleine-III showed an undefined conformational preference in the presence of SDS, indicating that the decreased overall charge of jelleine-III (+1) impairs both the electrostatic peptide–membrane interaction and biological activity [18]. 

Therefore, changes in the *C*- and *N*-terminal portions of the jelleine-I may influence its therapeutic efficacy [13,18]. In this sense, Romanelli et al. [18] found that the synthetic addition of a triad formed by two glycine residues and one tyrosine residue (YGG) in the *N*-terminal region of the jelleines I–III contributes to the reduction or even promotes the total loss of their biological activity. However, the incorporation of the same triad of amino acid residues into the *C*-terminal portion of jelleines not only did not alter their pharmacological activity but also increased the activity of the natural compound against some bacteria such as *Staphylococcus aureus*, *Escherichia coli* and *Salmonella enterica* subs. *enterica* serov. Paratyphi. 

Dynamic molecular simulations have shown equivalent solvation profiles of the *C*-terminal region for all four natural jelleines [50]. Nevertheless, the lowest degree of solvation of the *N*-terminal region of jelleine-I is observed when compared to other jelleines, indicating high hydrophobicity in this part of jelleine-I [50]. In fact, the presence of the Thr^1^ and Glu^1^ at position one of the jelleine-II and jelleine-IV sequences, respectively, attracts more water molecules to the *N*-terminal of these peptides, which affects their ability to interact with membrane interfaces [50]. On the other hand, the presence of the nonpolar Pro^1^ in jelleine-I allows high binding energy with the SDS micelle interface, highlighting the importance of the *N*-terminal region of this octapeptide in determining its biological effects. 

In addition, the His^7^ residue was also shown to have low binding energy at the membrane, suggesting that the initial interaction with Pro^1^ allows for the proper positioning of His^7^ for binding to the membrane. Interestingly, conductivity assays in single channels showed that the jelleine-I induces a higher electric current on bacterial membranes at pH 5.5 compared to pH 7.2, revealing the role of the protonated state of His^7^ in driving the electrostatic peptide-membrane interaction [50]. Consequently, the pore-forming activity of jelleine-I is improved under acidic conditions, as observed for the most cationic antimicrobial peptides [50]. Since many clinical conditions such as inflammation, infection and cancer are associated with low pH, these findings point to a value selectivity of jelleines as their biological actions are favored in dysfunctional environments over homeostatic environments.

Romanelli et al. [18] developed analogues from jelleines by inserting a Tyr residue with an ultraviolet probe and a spacer composed of two Gly residues coupled in the *C*-terminal (GGY; RJ I-IIIC) or *N*-terminal (YGG; RJ I-IIIN) region of jelleines I–III (Table 1). Moreover, Jia et al. [28] added halogen atoms (iodine, I-J-I; chlorine, Cl-J-I; bromine, Br-J-I; fluor, F-J-I) to the aromatic ring of the phenylalanine (Phe) amino acid residue, present in position 2 of jelleine-I. The halogen was added at the para position (carbon 4 of the ring) to avoid possible steric hindrance with the side chain of the Phe (Table 1). Thus, based on these two pioneering studies, several other authors have proposed modifications to jelleines. These chemical modifications aim to optimise the therapeutic activities of these compound groups by reducing their toxic effects and/or improving their plasmatic stability. The chemical strategies currently employed and the effects of these modifications on the activity, safety, stability and pharmacokinetics of jelleines are described in detail in the following sections of this review.

## 4. Pharmacological Proprieties

### 4.1. Antibacterial Activity

Antimicrobial resistance is an important threat to public health, and the development of new therapeutic agents is critical to address this problem [51]. For example, in 2019, there were an estimated 4.95 million deaths related to antimicrobial resistance, of which 1.3 million were directly attributable to infections caused by multidrug-resistant (MDR) bacteria [52]. This number is expected to reach 10 million deaths per year worldwide by 2050 [53], if not sooner, partly due to the widespread over-prescription of antimicrobials to COVID-19 patients in recent years [54,55]. Compared to conventional antimicrobials, AMPs possess certain advantages, such as high permeabilisation and internalisation in microbial cell membranes and, consequently, lower likelihood of developing bacterial MDR pathogens, broad-spectrum antibacterial activity, lower effect of accumulation in tissues, ability to neutralise virulence, lack of generation of active waste that can contaminate the environment and ability to modulate the host immune response [5,56,57]. 

Cationic AMPs generally have 12–50 amino acids with a net positive charge (between 2+ to 7+) as a consequence of excess of basic amino acid residues (i.e., lysine, arginine and histidine), and more than 50% in hydrophobic residues composition (i.e., glycine, alanine, proline, valine, leucine, isoleucine and methionine). These structural features are directly associated with the interactions with biological membranes, allowing electrostatic interaction with the anionic surface of microbial membranes and ensuring the hydrophobic interaction with the inner part of the membranes [58,59]. Jelleines consist of 8–9 amino acid residues with a hypothetical net charge of 2+ or 3+ and most of these residues are hydrophobic (38–50%), which is consistent with other known AMPs [59].

The antibacterial activity of jelleines and analogues has been widely investigated [13,18,19,20,21,22,23,24,25,26,27,28]. A biomonitoring study conducted by Fontana et al. [13] shows the antibacterial activity as a sensor to identify the fractions in which naturally occurring jelleines are presented. The advantage of studying the antibacterial activity of natural peptides such as the jelleine family is the potential to overcome antimicrobial resistance [60]. 

Several authors have shown that jelleine-I is active against gram-positive (*Staphylococcus aureus*, *Staphylococcus saprophyticus*, *Staphylococcus epidermidis*, *Streptococcus pneumoniae*, *Enterococcus faecalis*, *Bacillus subtilis*, *Bacillus cereus* and *Listeria monocytogenes*) and gram-negative (*Escherichia coli*, *Enterobacter cloacae*, *Klebsiella pneumoniae*, *Pseudomonas aeruginosa*, *Salmonella enterica* subs. *enterica* serovar Paratyphi, *Fusarium nucleatum* and *Citrobacter sakazakii*) bacteria of medical interest (Table 2) [13,18,19,20,21,22,23,24,25,26,27,28]. Jelleine-I was also effective against clinical isolates of MDR bacteria such as methicillin-resistant *S. aureus* (MRSA; MIC 128 µg/mL) [23], extended-spectrum beta-lactamase (ESBL)-producing *E. coli* (MIC 32 µg/mL) [23], beta-lactam-resistant *S. epidermidis* (MIC 64 µg/mL) [29] and piperacillin-resistant *P. aeruginosa* (MIC 32 µg/mL) [23]. Furthermore, isolates of *S. aureus* and *E. coli* resistant to jelleine-I were not recovered after 21 days of passage in a medium containing subinhibitory concentrations of this peptide, revealing the low potential of the jelleine-I to induce bacterial resistance in vitro [23].

Jelleine-I also showed efficacy in the in vivo infection model composed of *Galleria mellonella* larvae inoculated with 3 × 10^7^ colony-forming units (CFU)/mL of *L. monocytogenes*. Jelleine-I was able to, at 10 mg/kg, 20 mg/kg and 40 mg/kg, induce increased invertebrate survival by 10%, 20% and 30%, respectively. In addition, this peptide improved the number of immune cells in the larvae (haemocytes), reducing both the production of pro-inflammatory cytokines (TNF-α and IL-6) and the bacterial load in a dose- and time-dependent manner [24].

Jelleine-II showed a very similar spectrum of activity as jelleine-I, except for the lack of activity against *S.* Paratyphi and *L. monocytogenes* and the low activity against *E. coli* (MIC of 15 µg/mL for jelleine-II versus MIC of 2.5 µg/mL for jelleine-I) [13,18]. On the other hand, jelleine-III was also active, but with a narrower antibacterial spectrum, covering only *S. aureus*, *S. saprophyticus*, *E. coli* and *P. aeruginosa* [13]. Anotherstudy by Arpornsuwan et al. [61] showed that both jelleine-I and jelleine-II had low antibacterial activity against carbapenem-sensitive and carbapenem-resistant strains of *K. pneumoniae*, *E. coli*, *E. cloacae* and *C. freundii*. However, it is worth noting that the authors assessed antibacterial activity using the colorimetric 3-(4,5-Dimethylthiazol-2-yl)-2,5-diphenyltetrazolium bromide (MTT) method, which is not recommended in official guidance documents, such as CLSI [62] and EUCAST [63], since they are unable to estimate the biological effect of bacteriostatic concentrations of compounds with antibacterial activity. The antibacterial activity of jelleine-I was optimised by the synthesising of some analogues (Table 2). For example, the addition of one tyrosine and two glycines (YGG sequence) to the *C*-terminal region of jelleine-I increased the activity of this peptide against clinical isolates of *S. aureus* (MIC from 200 µg/mL to 100 µg/mL), *E. coli* (MIC from >200 µg/mL to 30 µg/mL) and *S.* Paratyphi (MIC from 200 µg/mL to 80 µg/mL) [18]. In addition, good antibacterial activity was observed for this jelleine-I analogue against ten clinical MDR isolates of *S. epidermidis* [29]. 

Jia et al. [28] reported halogenated derivatives of jelleine-I (i.e., F-J-I, Cl-J-I, Br-J-I and I-J-I) with 2–8-fold higher antimicrobial activities than those of the natural peptide. The presence of halogens, especially chlorine, bromine and iodine, increased the binding affinity of jelleine-I to bacterial lipopolysaccharide, justifying the improved activity of these analogues. In addition, the use of halogenated analogues of jelleine-I more efficiently protected mice infected with *E. coli* (0.1 mL of a bacterial suspension containing 3 × 10^8^ CFU/mL). The survival rate of mice receiving jelleine-I (20 mg/Kg) was 25%, while the survival rates in the group treated with F-J-I, Cl-J-I, Br-J-I and I-J-I (all analogues were also at 20 mg/Kg) were 50%, 62.5%, 62.5% and 37.5%, respectively [28]. 

The conjugation of jelleine-I with thiolated chitosans via the disulphide bond was also associated with an increase in the antimicrobial activity of the peptide against *S. aureus*. This chemical modification reduced the activity concentrations of jelleine-I from 96 µg/mL to 2.8–6.0 µg/mL [27]. However, Zahedifard et al. [30] showed that the conjugation of lauric acid, a saturated fatty acid, to the *N*-terminal region of jelleine-I resulted in a complete loss of activity against *E. coli*. As indicated by molecular dynamics studies [50], this region is essential for interaction with biological membranes, which could explain the loss of activity after the addition of lauric acid in the study by Zahedifard et al. [30]. Alternatively, the binding of this fatty acid to the *C*-terminal region may not promote a major impact on the biological activity, although future studies should be conducted to test this hypothesis.

AMPs can act through different pathways. The best-known effect of these agents is their binding to the anionic membranes of pathogens, destabilising this structure. Studies using scanning electron microscopy confirmed the pore-forming ability of jelleine-I in the bacterial membrane of *S. aureus*, *S.* Paratyphi and *L. monocytogenes* [18]. Jelleine-I acts on *S. aureus* cells and causes asymmetric divisions, pores in the membrane and leakage of cell contents. Moreover, this peptide leads to a considerable shortening of the bacilli and the appearance of invaginations in the *Salmonella* membrane, indicating pore formation [18]. In turn, *L. monocytogenes* cells treated with jelleine-I exhibit irregular morphology and cellular damage, including membrane shrinkage, depressions and empty cells, membrane pores, as well as surface roughening and deformation [24]. Therefore, the primary mechanism of action of jelleines is based on the disruption process in the bacterial cell membrane.

Currently, five different membrane-target models are considered to explain the mode of action of AMPs on biological membranes [64,65]: (i) the Barrel-stave model, in which the peptide monomers associate and form a bundle of helices embedded in the membrane, forming a transmembrane channel [66]; (ii) the toroidal pore model (also called wormhole), in which the amphipathic peptide interacts electrostatically with the phospholipid–plasma membrane in a perpendicular orientation relative to the membrane bilayer. In this way, the peptide exerts pressure on the membrane and separates the polar heads elements of phospholipids, resulting in the upper lipid monolayer curving bend through the pore in such a way that the pore lumen consists of peptide molecules and polar heads of phospholipids mixed together [66]; (iii) the carpet-like model, in which the AMP accumulates on the surface of the plasmatic membrane due to the electrostatic interaction between the positively charged peptide residues and the negative charges of phospholipids, covering it similarly to a carpet, which lead to membrane permeabilisation [65,66]; (iv) the aggregate channel model, which assumes that a supramolecular peptide–lipid complex is formed that mediates the mutually coupled trans-bilayer transport of lipid and peptides, implicitly assuming that informal aqueous channels exist within these aggregates, allowing the free passage of ions and possibly larger molecules [64,67]; and (v) the detergent-like model, in which AMPs aggregates in the membrane surface and after reach a critical concentration promote a micellisation process in the phospholipid bilayer [65]. Molecular dynamics simulations showed that jelleine-I acts by increasing the pressure on the lipid bilayer of the bacterial membrane [50]. This peptide accumulates in the outer membrane surface through electrostatic interaction between positively charged peptides and negatively charged phospholipid headgroups before significant leakage occurs. After the aggregation process, jelleine-I exerts pressure in the bilayer in order to better accommodate the polar and nonpolar residues in the amphiphilic environment of the membrane, resulting in the formation of a toroidal pore.

Interestingly, the jelleine-I analogue containing the YGG tail attached to the *C*-terminus shows a different mechanism of bacterial cell lysis. However, in the presence of SDS micelles, this peptide adopts a β-sheet conformation, and the jelleine-I peptide presents an α-helix under similar conditions. Although the YGG-jelleine-I also forms aggregates when interacting with the phospholipid membrane, the altered secondary structure permeabilises this structure by a mechanism based on a carpet-like model [18].

Although the primary mechanism of jelleines antibacterial action involves binding to the plasma membrane, several studies have identified possible non-target membrane models (Figure 3). Some authors showed that jelleine-I and its analogues induce the formation of reactive oxygen species (ROS) in *E. coli* and *S. aureus* [21]. Jia et al. [21] demonstrated that jelleine-I inhibits the electrophoretic motilities of DNA, suggesting this peptide could bind to genomic DNA and affect the genetic replication of microorganisms. Furthermore, these authors showed that the intracellular ATP content of *E. coli* can be reduced by up to 31% by jelleine-I, affecting the energy metabolism of the bacteria. 

Another strategic approach to combat MDR bacteria is to explore synergism by combining two or more antimicrobial agents in order to promote improvements in antimicrobial properties, economic costs and side effects of antimicrobial therapy [68,69,70,71]. One of the most common methods to assess the synergism from the combination of different compounds is to determine the fractional inhibitory concentration index (FICI), which indicates a synergistic effect when the value is less than 0.5 [56,72]. The antimicrobial activity of jelleine-I was evaluated when co-administered with AMPs secreted by the granule glands of the European red frog (*Rana temporaria*), the so-called temporins. A synergistic effect was observed after its combination with temporin A (FLPLIGRVLSGIL; FIC 0.3) and temporin B (LLPIVGNLLKSLL; FIC 0.4) against *S. aureus* A170 and *L. monocytogenes*, respectively [18]. Furthermore, Capparelli et al. [29] showed that the combination of the jelleine-I analogue containing YGG at the *C*-terminus, with an analogue of Temporin TB (TB-KK; KKYLLPIVGNLLKSLL-NH_2_), exhibited a strong synergistic effect (FICI ≤ 0.5). The combination (GGY-J-I plus TB-KK), when injected intravenously 3 h after the infection of mice with *S. epidermidis* (10^8^ CFU/mouse), was able to reduce the bacterial load in the kidneys and spleen of the animals to a similar level as the gentamicin-treated groups. In addition, the levels of acute phase reaction marker proteins (i.e., serum amyloid A, haptoglobin and fibrinogen) were within the normal range in the mice treated with the combination, and the expression of TNF-α, IFN-γ and COX-2 genes in the kidney was down-regulated in these animals, but not in the group that received gentamicin. In addition, the combination was shown to reduce the number of granulocytic cells in the cortical region of the kidney of the infected animals. After eleven days, the mice treated with gentamicin were still infected, while the animals treated with the peptide mixture were already sterile [29]. These results show that jelleine-I is a promising prototype for the development of new antibacterial therapies. Moreover, its use in monotherapy or combination therapy could lead not only to the control of infection but also to the reduction of inflammation associated with this problem.

### 4.2. Antifungal Activity

Fungi are increasingly recognised as the cause not only of superficial infections that affect many people and are relatively easy to treat but also of invasive and disseminated infections [73]. Fungal spores contribute to significant reactive respiratory disease in more than 10 million people. In addition, it is estimated that fungal diseases affect more than one billion people worldwide, of whom more than 300 million people suffer from severe fungal infections each year, resulting in 1.5 million deaths [74,75]. Nowadays, only four classes of antifungal agents are available for the treatment of severe invasive infections (i.e., azoles, polyenes, echinocandins and flucytosine), and resistance to these agents threatens the efficacy of currently available therapy [3,76,77,78]. The high mortality rate of invasive fungal infections, long treatment duration, narrow spectrum of activity and cross-resistance due to similar mechanisms of action among all drugs have therefore triggered the search for safer alternatives with lower toxicity or other improved properties [79]. In this context, AMPs are also promising alternatives to expand the narrow antifungal therapeutic arsenal currently available [80], and jelleines have shown significant antifungal properties.

Fontana et al. [13] showed high activity for jelleine-I and jelleine-II against *Candida albicans* (MIC 2.5 µg/mL), whereas Kim et al. [25] reported higher MIC (16 µg/mL) against the same yeast. Jia et al. [20], in turn, confirmed that jelleine-I is also active against non-albicans *Candida* species (*C. tropicalis*, *C. parapsilosis*, and *C. glabrata*), showing an extended spectrum against this genus of pathogenic yeasts (Table 3). The antifungal activity was predominantly fungicidal, and jelleine-I was able to produce maximum microbicidal activity after 3 h of exposure. Electron microscopic examination showed that the membrane surfaces of *C. albicans* and *C. glabrata* cells were distended and rough after treatment with jelleine-I. Thus, this result shows that jelleine-I acts on the fungal membrane and causes significant damage that promotes microbial lysis, which was confirmed by the release of 260 nm absorbance material and propidium iodide assays [20]. In addition, jelleine-I has been shown to bind to polysaccharides of the fungal cell wall (i.e., laminarin and mannans) and stimulate the formation of reactive oxygen species (ROS), favoring its biological action against *Candida* [20]. In turn, jelleine-III and jelleine-IV did not show any activity against *C. albicans* (Table 3), suggesting low antifungal activity in these derivatives.

Treatment of animals with candidemia with jelleine-I resulted in reduced mortality. Kunming mice were infected intravenously with *C. albicans* (0.1 mL saline contained 2 × 10^6^ CFU/mL). One hour after infection, the animals were injected intraperitoneally with different doses of jelleine-I (0.5, 1, 5 and 10 mg/Kg) once daily for 7 days. At the end of the 14-day experiment, the mortality of the animals ranged from 100% in the untreated group, while only 40% in the group treated with jelleine-I (10 mg/Kg). The antifungal effect observed with jelleine-I was superior to that of fluconazole, which reduced mortality to 60% of the untreated control (5 mg/Kg) [20].

### 4.3. Antiparasitic Activity

Another important public health challenge is related to the infections associated with parasitic pathogens such as *Leishmania*, which causes a vector-borne disease endemic in 98 countries [81]. Leishmaniases are zoonoses manifested in cutaneous, mucocutaneous and visceral forms [81,82,83]. Among all parasitic infections, leishmaniasis is the second most lethal, with 20,000–30,000 deaths per year [81]. The therapeutic arsenal used for leishmaniasis consists mainly of pentavalent antimony, paromomycin, pentamidine, miltefosine and amphotericin B [84]. Nevertheless, these drugs are both toxic and prone to side effects, and are not available to poor populations due to high manufacturing costs [82]. Accordingly, new drug candidates against leishmaniasis are urgently needed and AMPs have shown potential for the therapeutic treatment of this parasitosis [85].

Jelleine-I presents low anti-leishmania activity, being able to affect the multiplication of *L. major* promastigote with an IC_50_ value of 400 µg/mL. However, *L. major* amastigotes in inside macrophages (THP_1_ cells) were not susceptible to the antiparasitic effects of jelleine-I [30]. Investigation of the mechanism of action of jelleine-I on *L. major* amastigotes revealed that this peptide induces a rapid action in the membrane of the pathogen by changing electrical potential and increasing membrane permeability, which contributes to parasite cell collapse. Scanning electron microscopy of promastigotes of *L. major* exposed to jelleine-I (400 µg/mL) confirms this effect on the membrane. Notably, a round shape disfiguring promastigotes from their cylindrical structures is observed, while the presence of surface vesicles is noted in parasites that retained their normal shape [30]. However, the effect of jelleine in combination with antileishmanial agents remains to be investigated, as does the antiparasitic potential of synthetic analogues of this natural peptide.

### 4.4. Anti-Inflammatory Effect

Inflammation is a physiological mechanism by which organisms defend themselves against infection and restore homeostasis in damaged tissue [86]. However, if the resolution of the inflammatory process does not occur appropriately, several health problems may be observed, such as tissue damage with loss of function, fever, pain and edema [87]. Treatment of inflammatory diseases employs small molecules as drugs to interact with a large number of pharmacological targets [88]. Unfortunately, they are highly toxic, poorly selective and associated with a variety of undesirable side effects, including the breakdown of the blood brain barrier and the generation of toxic molecules after the metabolic process; they are often unsuitable for long-term therapy [86,88]. Therefore, an alternative treatment based on the use of bioactive peptides as anti-inflammatory agents is currently being developed, including jelleines as a promising prototype. 

Jelleine-I reduces the production of pro-inflammatory cytokines by macrophages in vitro. Jia et al. [21] showed that the concentration of TNF-α in the culture medium of LPS-stimulated murine RAW264.7 cells was 655.81 ± 8.27pg/mL, while treatment with 84 μg/mL of jelleine-I resulted in a concentration of TNF-α of 370.47 ± 28.97 pg/mL [21]. Another anti-inflammatory study employing the combination of the analogue RJI-C (YGG-Jelleine-I) with temporin TB (RJI-C 9 μg/mL + TB-KK 6 μg/mL) revealed the inhibition of the TNF-α and IFN-γ synthesis in J774 cells (10^6^ cells/well) stimulated with LPS for 3 h to the same extent as acetylsalicylic acid (5 μg/mL) as well as the level of COX-2 in these cells [29]. In addition, in mice stimulated with LPS (250 μg, ~10 mg/Kg), the peptide mixture decreased the levels of pro-inflammatory (TNF-α and IFN-γ) and anti-inflammatory (IL-10) cytokines [29]. However, little is known about the anti-inflammatory molecular mechanisms of jelleine-I and the role of this peptide in complications related to the inflammatory process. Particularly, the effect of jelleine-I in pain control needs to be further investigated. 

### 4.5. Healing and Hemostatic Effect

Wound healing refers to the complex and multifactorial process in response to a disruption of the normal structure of an epithelial tissue [89,90,91,92]. The attempt to restore the lesion caused by local aggression begins with an inflammatory and coagulative phase leading to repair consisting of the replacement of specialised structures by the deposition of collagen and ends with the regenerative phase characterised by the process of tissue re-epithelialisation [91]. However, wounds that exhibit impaired healing, including delayed acute wounds and chronic wounds, generally have not progressed through the normal healing phases [90,92]. In these cases, the wound healing process can be promoted by products with medicinal properties, such as natural peptides [89,93]. In this way, many studies have been conducted on the wound-healing properties of natural peptides with anti-inflammatory, antioxidant, antibacterial and collagen-synthesis-promoting effects. 

Although there is no evidence for its healing effect in isolated form, jelleine-I in combination with halogenated adenosines promotes strong tissue regeneration. Zhou et al. [31] showed that a biocompatible and biodegradable hydrogel containing jelleine-I (6 mM) and 8-bromoadenosine-3′,5′-cyclic monophosphate, a derivative of cyclic monophosphate, induced significant wound healing in mice with a diabetic wound infected with MRSA. The wound in the hydrogel-treated animals decreased significantly at days 7 and 14 compared to untreated animals. In the histological analysis, the epidermis of the wound in the hydrogel-treated group recovered almost completely, and the densities of blood vessels, hair follicles, muscle layer, granulation tissue and other skin appendages were much better than observed in untreated animals. These findings revealed the hydrogel containing jelleine-I is efficient in accelerating wound healing and shortening the wound healing time [31]. Furthermore, the hydrogel showed excellent antimicrobial activity against MRSA and induced the secretion of tissue (TGF-β) and vascular (VEGFA) growth factors in vitro and in vivo [31].

In addition, jelleine-I also helps to stem bleeding that follows tissue injury. In combination with the platelet-aggregating adenosine diphosphate, jelleine-I reduced the release of hemoglobin from a clot formed in vitro and resulted in a complete stop of bleeding after 23 s of exposure to liver samples from mice with acute hemorrhage induced by puncture [32]. The antimicrobial effect combined with the angiogenic, proliferative and hemostatic activity makes jelleine-I a promising compound for the development of new treatments for acute and chronic wounds.

### 4.6. Antitumoral Activity

Cancers are a group of diseases characterised by uncontrolled growth and spread of abnormal cells. If left unchecked, the spread of cancer cells at this stage, known as metastasis, can lead to death [94]. It is estimated that the global cancer burden will increase to 20 million new cases and 10 million deaths by 2020 [95]. In this scenario, 1 in 5 men and 1 in 6 women worldwide will develop cancer in their lifetime, with 1 in 8 men and 1 in 11 women dying from the disease [95]. Although many effective anticancer drugs are available, the majority of drugs currently used are not specific, raising issues such as the frequent side effects associated with cancer chemotherapy and widely documented multidrug resistance [96,97,98]. In order to reduce the negative effects of chemotherapy, the development of new antitumor drugs is essential, and natural peptides have proven to be excellent strategies for this purpose [99,100].

Jelleine-I and its halogenated analogues (J-I-I, J-I-Cl, J-I-Br, J-I-F) show value potential to be used as adjuvants in the treatment of colorectal cancer (CRC), the second most common type of cancer worldwide. These peptides inhibit *Fusobacterium nucleatum*, an anaerobic bacterium of the oral microbiota that is highly active in the altered microecology of the gut and is closely associated with the initiation and progression of CRC. Compared to jelleine-I, whose MIC is 160 μg/mL, the MICs of Br-J-I, Cl-J-I and I-J-I were 5 μg/mL, 10 μg/mL and 10 μg/mL, respectively, representing at least a 10-fold improved potential against *F. nucleatum* [22]. The brominated analogue (J-I-Br) increased the formation of ROS and caused damage to the membrane of *F. nucleatum*, in addition to strong binding to *Fusobacterium* adhesin A (FadA). FadA is located in the outer membrane of *F. nucleatum* and molecular docking studies suggest that Br-J-I induces oligomerisation of FadA in the membrane to permeabilise this structure [22]. Br-J-I showed no significant effect on the viability of colon cancer cells HCT116, Lovo, HT29 and MC38 in vitro only at a high concentration (80 µg/mL), but this jelleine-I analogue inhibits cell proliferation of human colon cancer cells (HCT116) induced by *F. nucleatum* in culture and in animal models [22]. In mice bearing HCT116 xenografts infected with *F. nucleatum*, the application of Br-J-I decreased tumor cell proliferation (as indicated by decreased Ki-67 content), tumor weight, inflammation in the microenvironment of CRC cells (decreased TNF-α and IL-1β) and increased expression of cell adhesion molecules (occludin and zonula occludin-1) [22]. In addition, the combination of Br-J-I with 5-fluoruracil, one of the most commonly prescribed drugs in the therapy of CRC, showed a strong synergistic effect, suggesting its use as a combination therapy [22].

## 5. Toxicity

Despite their potent antibacterial effects, one of the main limitations of using AMPs is related to their toxicity [4,5,56]. This is because many of these compounds also act on the zwitterionic cell membrane characteristics of mammalian cells, resulting in significant cytotoxicity [101]. This is particularly common in kidney and liver cells, which tend to accumulate AMPs during excretion and metabolism, respectively [102]. Therefore, characterising the toxicity of biologically active AMPs is essential for a more robust assessment of their therapeutic potential. Several studies have shown very low toxicity for the natural jelleines [13,20,21,24] and their synthetic analogues [18,28]; therefore, they are suitable as safe prototypes for the development of new pharmacological agents.

Fontana et al. [13] showed that natural jelleines have low hemolytic activity and do not induce mast cell degranulation, indicating low allergic potential. The release of 5(6)-Carboxyfluorescein (CF) from phospholipid bilayers showed that jelleine-I was more responsive to anionic vesicles—L-α-phosphatidylcholine:L-α-phosphatidylglycerol sodium salt (70:30), which mimics a general bacterial membrane, and L-α-phosphatidylcholine:L-α-phosphatidyl-L-serine:ergosterol (40:40:20), which mimics a *C. albicans* membrane, when compared to zwitterionic micelles—L-α-phosphatidylcholine:L-α-phosphatidylcholine:cholesterol (40:40:20)—mimicking an average red blood cell [50]. Similarly, Jia et al. [20] showed that no significant hemolytic effect was observed with jelleine-I even at very high concentrations (256 µg/mL), whereas Zahedifard et al. [30] found only 8% hemolysis with jelleine-I at 500 µg/mL, which was completely lost with the analogues of this peptide conjugated to fatty acids (lauric acid).

The cytotoxicity of jelleine-I was also very low, underlining its low toxicity. At a concentration of 100 µg/mL, this peptide caused only a 14% loss of viability in cells of the kidney of an African green monkey (*Chlorocebus sabaeus*; Vero lineage) and only 5% lysis in human red blood cells [61]. Otherwise, in mouse embryo fibroblasts (NIH 3T3 lineage), the jelleine-I (256 µg/mL) maintained cell culture viability at more than 80%, and toxicity was not altered by the addition of halogens (F, Cl, I and Br) in the phenylalanine residue of this compound [28]. Jelleine-I also had no effect on the viability of mammalian macrophages (THP_1_ lineage), with complete viability observed even in cells exposed to 250 µg/mL of the peptide [30]. Zhou et al. [23] and Jia et al. [22] showed that the viability of NIH 3T3, RAW264.7 and Hela cells was more than 85–90% after exposure to jelleine-I at 256 µg/mL, confirming the low in vitro toxicity.

Even in combination with other AMPs, jelleines have low toxicity. In this sense, the mixture of RJI-C with temporin TB (RJI-C 9 μg/mL + TB-KK 6 μg/mL) lysed less than 12% of the murine erythrocytes (HC_50_ 143.8 µg/mL) and was also not toxic to macrophages (J774 cells), which were still vital after 72 h [29]. Furthermore, the combination has little effect on probiotic bacteria (*Lactobacillus plantarum*, *Lactobacillus Paracasei*, *Bifidus animalis*), whereas gentamycin kills all of them [29]. This effect is relevant because many of the adverse drug reactions are related to their activity on the gut microbiota, which can lead to diarrhea, nausea, vomiting and even chronic changes in the gut mucosa [103].

The acute in vivo toxicity of jelleine-I confirmed the low toxicity of this peptide, with LD_50_ in healthy Kunming mice being >1000 mg/Kg. Jia et al. [20] revealed that in the group with the highest dosage of jelleine-I (1000 mg/kg), only one mouse died (each group contained 12 mice with six males and six females) after 14 days and that no obvious behavioral abnormalities were observed after administration of this peptide.

## 6. Stability

AMPs have multiple sites that are susceptible to hydrolysis by serum and tissue peptidases, which can significantly affect their stability [5,56,104]. Indeed, jelleine-I is highly susceptible to biological peptidases and loses its antimicrobial activity after incubation with trypsin or chymotrypsin at 1 × 10^−4^ mg/mL for 6 h. However, the addition of Cl, Br or I to the second amino acid residue of this peptide (phenylalanine) increases the proteolytic stability towards trypsin and chymotrypsin 10–100-fold. In blood, jelleine-I is somewhat more stable, with about 10% intact peptide remaining after a 2 h incubation in murine serum at 37 °C. In this case, the addition of Br or I to the peptide significantly increases serum stability, with about 50% intact peptides after 2 h of incubation under the same conditions [28]. Jelleine-I is able to form aggregates on biological membranes that can protect it from enzymatic degradation [50]. This behaviour may help us to understand the difference in serum stability compared to direct exposure to hydrolytic enzymes, as the former often contains substantial amounts of membrane fragments derived from erythrocytes that are ruptured during centrifugation, allowing the jelleine-I to form its aggregates.

## 7. Conclusions Remakes

Jelleines consist of a family of small peptides with multiple pharmacological actions reported in the literature. This review correlates some structural features with the most promising actions of jelleines (I–III). The *N*-terminal of jelleine-I has an important role in anchoring this short peptide properly driving the single histidine residue to bind to biological membranes, thereby promoting membrane permeabilisation. Therefore, changes in the *N*-terminal regions, as noted for jelleine-II and jelleine-III, interfere with the membrane interaction and biological properties. In addition, the leucine residue in the carboxyamide *C*-terminal is mandatory for the pharmacological activities of jelleines, since any antimicrobial activity is observed for jelleine-IV. 

Although the antibacterial activity of jelleine-I has been extensively studied, there is little evidence for the other therapeutic uses of this compound. The antifungal activity has only been established against pathogens of the *Candida* genus and should be further explored against filamentous and dimorphic species. In addition, the mechanisms involved in the anti-inflammatory effect and the analgesic potential of jelleine-I remain to be characterised. The use of jelleine-I in combination with antitumoral and antiparasitic drugs should also be better explored, as the very low toxicity of this natural peptide can be used to reduce the significant side effects associated with drugs. 

Furthermore, the pharmacological properties of other bioactive jelleines, i.e., jelleine-II and jelleine-III, need to be better evaluated. Currently, all studies that explore pharmacological activities other than antimicrobial have included only the jelleine-I. Moreover, the toxicological properties and the stability of these analogues need to be characterised in new studies.

It is worth noting, however, that jelleine-I in its native form should only be used for topical or local applications, as it is rapidly degraded by the systemic route. If intravenous use is required, chemical strategies should be promoted to increase its plasma stability, such as the addition of halogens, fatty acid binding, PEGylation, carbohydrate binding, serum protein binding or incorporation into nanoparticles or liposomes. In fact, any structural changes in the *N*-terminal region of the molecule should be avoided once it is part of the pharmacophore of jelleine-I. In summary, jelleine-I is a promising prototype for new drug development due to its low toxicity, small size and good antibacterial activity.

## Figures and Tables

**Figure 1 toxins-16-00024-f001:**
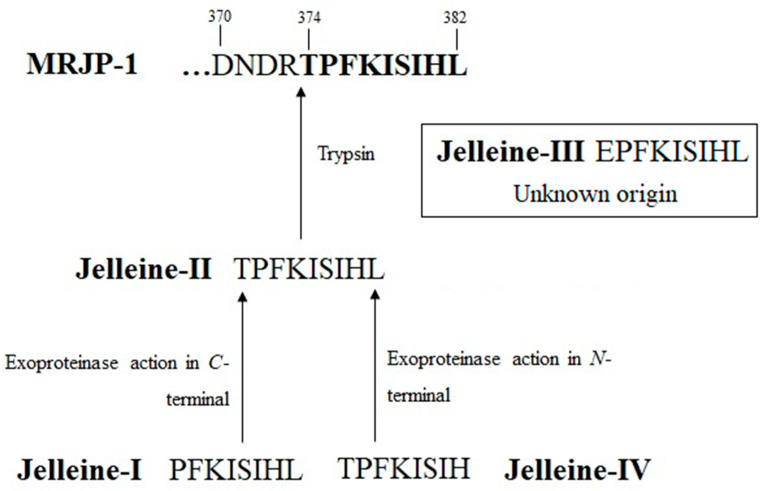
Schematic representation of the production of jelleines from Major Royal Jelly Proteins (MRJP)-1. As shown, jelleine-II is obtained via hydrolysis of MRJP-1 by the trypsin present in royal jelly. Subsequently, unknown exoproteinases act on the *N*-terminal and *C*-terminal positions of the jelleine-II, resulting in the formation of the jelleine-I and jelleine-IV, respectively. The origin of jelleine-III is currently unknown. The quotation mark indicate that the sequence of the protein amino acid residues continues in the *N*-terminal direction.

**Figure 3 toxins-16-00024-f003:**
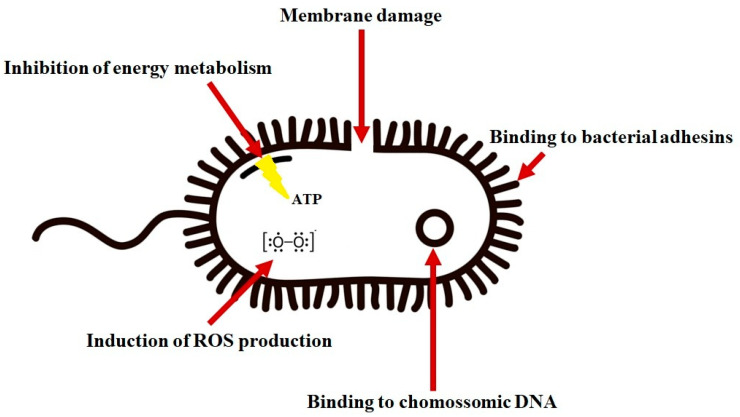
The major biochemical mechanisms of jelleine-I on bacterial cells. Reactive Oxygen Species—ROS; Adenosine Triphosphate—ATP. Figure made by the authors.

**Table 1 toxins-16-00024-t001:** Main chemical characteristics of natural jelleines and some of their main synthetic analogues.

Name	Sequence	Residues Number	Molecular Masses (Da)	Net Charge at pH 7	Basic Residues	Acid Residues	Aromatic Residue	Hydrophobic Residues (%)	References
Jelleine-I	**PFKISIHL-NH_2_**	8	953.24	2	2	0	1	50	[13]
Jelleine-II	**TPFKISIHL-NH_2_**	9	1054.30	2	2	0	1	44	[13]
Jelleine-III	**EPFKISIHL-NH_2_**	9	1082.32	1	2	1	1	44	[13]
Jelleine-IV	**TPFKISIH-NH_2_**	8	942.13	2	2	0	1	38	[13]
RJ IC	**PFKISIHLGGY-NH_2_**	11	1230.46	2	2	0	2	55	[18]
RJ IIC	**TPFKISIHLGGY-NH_2_**	12	1331.56	2	2	0	2	50	[18]
RJ IIIC	**EPFKISIHLGGY-NH_2_**	12	1359.57	1	2	1	2	50	[18]
RJ IN	**YGGPFKISIHL-NH_2_**	11	1230.46	2	2	0	2	55	[18]
RJ IIN	**YGGTPFKISIHL-NH_2_**	12	1331.56	2	2	0	2	50	[18]
RJ IIIN	**YGGEPFKISIHL-NH_2_**	12	1359.57	1	2	1	2	50	[18]
F-J-I ^A^	**PF^F^KISIHL-NH_2_**	8	972.99	2	2	0	1	50	[28]
Cl-J-I ^A^	**PF^Cl^KISIHL-NH_2_**	8	989.63	2	2	0	1	50	[28]
Br-J-I ^A^	**PF^Br^KISIHL-NH_2_**	8	1033.98	2	2	0	1	50	[28]
I-J-I ^A^	**PF^I^KISIHL-NH_2_**	8	1080.95	2	2	0	1	50	[28]

Yellow: hydrophobic residues; green: aromatic residues; red: Acid residues; blue: basic residues. ^A^ The halogen atoms were added directly linked to the aromatic ring of the phenylalanine residue of jelleine-I, all in para position. In the sequence presented in Table 1, halogens are represented by letters superscript to the phenylalanine residue (F). F: Fluorine; I: Iodine; Br: Bromine; Cl: Chlorine.

**Table 2 toxins-16-00024-t002:** Minimum inhibitory concentrations (MIC; µg/mL) of natural jelleines and their synthetic analogues against gram-positive and gram-negative bacteria of medical interest.

Peptide	Gram-Positive (MIC in µg/mL)								Gram-Negative (MIC in µg/mL)								Reference
	*S. aureus*	*S. saprophyticus*	*S. epidermidis*	*B. subtilis*	*B. cereus*	*L. monocytogenes*	*E. faecalis*	*S. pneumoniae*	*E. coli*	*E. cloacae*	*K. pneumoniae*	*P. mirabilis*	*P. aeruginosa*	*S.* Paratyphi	*C. sakazakii*	*F. nucleatum*	
Jelleine-I	8–128	15	64	4–32	I	12.5	16	6	2.5–32	10	8–64	I	8–62	200	64	160	[13,18,19,20,21,22,23,24,25,26,27,28]
Jelleine-II	15-I	10	-	30	I	I	-	-	15	15	15	I	15	I	-	-	[13,18]
Jelleine-III	30-I	30	-	I	I	I	-	-	15	I	I	I	30	I	-	-	[13,18]
Jelleine-IV	I	I	-	I	I	-	-	-	I	I	I	I	I	-	-	-	[13]
RJ IC	100	-	30	-	-	I	-	-	30	-	-	-	-	80	-	-	[18]
RJ IIC	I	-	200	-	-	I	-	-	I	-	-	-	-	I	-	-	[18]
RJ IIIC	I	-	300	-	-	I	-	-	I	-	-	-	-	I	-	-	[18]
RJ IN	NR ^A^	-	-	-	-	NR ^A^	-	-	NR ^A^	-	-	-	-	NR ^A^	-	-	[18]
RJ IIN	NR ^A^	-	-	-	-	NR ^A^	-	-	NR ^A^	-	-	-	-	NR ^A^	-	-	[18]
RJ IIIN	NR ^A^	-	-	-	-	NR ^A^	-	-	NR ^A^	-	-	-	-	NR ^A^	-	-	[18]
F-J-I	64	-	16	16	-	-	-	-	32	-	64	-	32	-	32	-	[28]
Cl-J-I	32	-	16	8	-	-	-	-	16	-	16	-	16	-	16	10	[22,28]
Br-J-I	32	-	16	8	-	-	-	-	16	-	16	-	16	-	16	5	[22,28]
I-J-I	32	-	8	8	-	-	-	-	16	-	16	-	16	-	8	10	[22,28]

I: Inactive; -: non-tested; NR: tested, but the MIC value was not reported. ^A^ Despite not reporting MIC values, the authors pointed out that a very low activity was detected against the bacteria for all the *N*-terminal-modified peptides.

**Table 3 toxins-16-00024-t003:** Minimum inhibitory concentrations (MIC; µg/mL) of natural Jelleines and their synthetic analogues against fungi of medical interest.

Peptide	Yeast					References
	*C. albicans*	*C. glabrata*	*C. tropicalis*	*C. krusei*	*C. parapsilosis*	
Jelleine-I	2.5–61	30	15	30	61	[13,20]
Jelleine-II	2.5	-	-	-	-	[20]
Jelleine-III	I	-	-	-	-	[20]
Jelleine-IV	I	-	-	-	-	[20]

I: Inactive; -: non-tested.

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
