# Peer review of "Jelleine, a Family of Peptides Isolated from the Royal Jelly of the Honey Bees (Apis mellifera), as a Promising Prototype for New Medicines: A Narrative Review"

_toxins, 2024, doi:10.3390/toxins16010024_

Round 1

Reviewer 1 Report

Comments and Suggestions for Authors

this study was about Jelleines, a family of peptides isolated from the royal jelly of the honey bees (Apis mellifera), as promising prototypes for new medicines.

This is an interesting and comprehensive review and highlights the importance of peptides isolated from, royal jelly showing a significant biological role.

It is recommended to publish with minor changes.

Royal jelly word can be added in keywords list

Table 3 format is not according to journal std. 

in Table 3 NT = abbreviation was used in footnote but there is no such word in the Table。 please double check this throughout the paper.

Reference: Please correct and modify reference (line 721)

Comments on the Quality of English Language

Minor editing of English language required.

Author Response

General comments: This study was about Jelleines, a family of peptides isolated from the royal jelly of the honey bees (Apis mellifera), as promising prototypes for new medicines.

This is an interesting and comprehensive review and highlights the importance of peptides isolated from, royal jelly showing a significant biological role.

It is recommended to publish with minor changes.

Answer: We would like to thank the reviewer for his suggestions, which certainly contributed to improving the quality of the manuscript. Below we have included a point-by-point response to each reviewer's suggestion. We appreciate your willingness to review our manuscript.

Question #1: Royal jelly word can be added in keywords list

Answer: Modification was performed as suggested by the reviewer.

Question #2: Table 3 format is not according to journal std. 

Answer: Modification was performed as suggested by the reviewer.

Question #3: in Table 3 NT = abbreviation was used in footnote but there is no such word in the Table。 please double check this throughout the paper.

Answer: Modification was performed as suggested by the reviewer.

Question #4: Reference: Please correct and modify reference (line 721)

Answer: Modification was performed as suggested by the reviewer.

Reviewer 2 Report

Comments and Suggestions for Authors

My comments :

Reformulate the abstract. Example: try to focus less on Jelleine I in the abstract insofar as there is no specific development of the Jelleine I part in the manuscript. 

List the references concerning the data presented in Table 1. 

Center the data in Table 1 with the titles presented in the same table. Explain in the Table I legend (or in the manuscript text) exactly what the analogs and halogens I, Cl, Br that appear as coefficients on certain amino acids in Table 1 correspond to. 

In Figure 1, show the sequence of Jelleine III! 

Quote the reference concerning the determination of the 3D structure of MRJP-I. 

On page 3, line 215, explain what SDS is.

In section 3 "Chemical properties of Jelleines" the authors have simply described the chemical properties of Jelleines I - IV only. The authors should also discuss the properties of other related analogous molecules, as they have also presented them in Table 1 (RJ IC, RJ IIC, RJ IIIC, RJ IN, RJ IIN, RJ IIIN, F-J-I, Cl-J-I, Br-J-I, I-J-I) !!!!!

In table 2, add a column to show the corresponding references.  

In the legend of table 3, quote the reference if the figure has been taken from the literature, or specify that the figure has been made by the authors if this is the case! 

In the legend to table 4, quote the corresponding reference. 

In section 4.2 "Antifungal activity", the authors should discuss Jelleins III and IV, since they have included them in the corresponding table 4.

Author Response

General answer: We would like to thank the reviewer for his suggestions, which certainly contributed to improving the quality of the manuscript. Below we have included a point-by-point response to each reviewer's suggestion. We appreciate your willingness to review our manuscript.

Question #1: Reformulate the abstract. Example: try to focus less on Jelleine I in the abstract insofar as there is no specific development of the Jelleine I part in the manuscript. 

Answer: Modification was performed as suggested by the reviewer.

Question #2: List the references concerning the data presented in Table 1.

Answer: Modification was performed as suggested by the reviewer.

Question #3: Center the data in Table 1 with the titles presented in the same table. Explain in the Table I legend (or in the manuscript text) exactly what the analogs and halogens I, Cl, Br that appear as coefficients on certain amino acids in Table 1 correspond to. 

Answer: Modification was performed as suggested by the reviewer.

Question #4: In Figure 1, show the sequence of Jelleine III! 

Answer: Modification was performed as suggested by the reviewer.

Question #5: Quote the reference concerning the determination of the 3D structure of MRJP-I. 

Answer: The citation was added and the study included in the list of references.

Question #6: On page 3, line 215, explain what SDS is.

Answer: The acronym was named in full the first time it was mentioned on page 2, line 195, in accordance with the periodical's rules. SDS stands for sodium dodecyl sulfate.

Question #7: In section 3 "Chemical properties of Jelleines" the authors have simply described the chemical properties of Jelleines I - IV only. The authors should also discuss the properties of other related analogous molecules, as they have also presented them in Table 1 (RJ IC, RJ IIC, RJ IIIC, RJ IN, RJ IIN, RJ IIIN, F-J-I, Cl-J-I, Br-J-I, I-J-I) !!!!!

Answer: Modification was performed as suggested by the reviewer.

Question #8: In table 2, add a column to show the corresponding references.

Answer: Modification was performed as suggested by the reviewer.

Question #9: In the legend of table 3, quote the reference if the figure has been taken from the literature, or specify that the figure has been made by the authors if this is the case! 

Answer: Information was added as suggested.

Question #10: In the legend to table 4, quote the corresponding reference. 

Answer: Information was added as suggested.

Question #11: In section 4.2 "Antifungal activity", the authors should discuss Jelleins III and IV, since they have included them in the corresponding table 4.

Answer: Information was added as suggested.

Reviewer 3 Report

Comments and Suggestions for Authors

Respected Authors,

Jelleines (I-IV) are a family of peptides that occur in royal jelly and may play an essential role in the treatment due to their biological activities. So the topic of the article seems interesting and valuable. They may be taken into consideration as potential components of new medicines. Jelleine I, II, III and IV are short peptides in royal jelly. In my opinion, the title should indicate that this article is a review. I would like to underline that the introduction is sufficient and well describes the topic. I suggest to correct the title of chapter 3. “Chemical proprieties of jelleines” as follows: chemical properties of jelleines. Moreover, the sentence in line 203-207 should be refreshed. In this part, the Authors should pay attention to the differences between structure jelleines I -III and jellein IV, due to the lack of antimicrobial effect of jellein IV.  This fact should be associated with structure and Authors should find the answer to this point. In the next chapter, Authors tried to explain the antimicrobial effect of

Next chapter 4 was dedicated to the pharmacological activity, however, it regards not only jellies but also other antimicrobial peptides, so I propose to mention about overview of the antimicrobial effect of royal jelly components. In subchapter 4.2. The authors described the wide antifungal effect of Jelleines I and II, but I couldn’t find information about III and IV. They have to be mentioned if they have this action or in literature is lack of information. A similar situation occurs in the case of the antifungal action of jelleines. Only in Table 3, we can find information that jelleines III and IV are inactive. This fact should be mentioned in the text also. Subchapters 4.3 and 4.4 as well as 4.5 ., and 4.6  are only information about jelleine 1.

The chapter 5 regarding toxicity focused on jelleine -I. I couldn’t find information about the rest proteins. I would like to highlight that the conclusions are well prepared and indicate on potential role of jelleine-I as a promising prototype for new drug.

The article lacks some formal information such as Author Contributions, Funding as well as Acknowledgments.

Comments on the Quality of English Language

minor corrections

Author Response

General comments: Jelleines (I-IV) are a family of peptides that occur in royal jelly and may play an essential role in the treatment due to their biological activities. So the topic of the article seems interesting and valuable. They may be taken into consideration as potential components of new medicines. Jelleine I, II, III and IV are short peptides in royal jelly.

Answer: We would like to thank the reviewer for his suggestions, which certainly contributed to improving the quality of the manuscript. Below we have included a point-by-point response to each reviewer's suggestion. We appreciate your willingness to review our manuscript.

Question #1: In my opinion, the title should indicate that this article is a review.

Answer: Modification was performed as suggested by the reviewer.

Question #2: I would like to underline that the introduction is sufficient and well describes the topic.

Answer: We thank the reviewer for the comment.

Question #3: I suggest to correct the title of chapter 3. “Chemical proprieties of jelleines” as follows: chemical properties of jelleines.

Answer: Modification was performed as suggested by the reviewer.

Question #4: Moreover, the sentence in line 203-207 should be refreshed. In this part, the Authors should pay attention to the differences between structure jelleines I -III and jellein IV, due to the lack of antimicrobial effect of jellein IV.  This fact should be associated with structure and Authors should find the answer to this point.

Answer: Modification was performed as suggested by the reviewer. As this structural issue is very important to justify the difference in activity observed between jelleines, we highlight it in the first paragraph of the section “chemical properties of jelleines”

Question #5: In the next chapter, Authors tried to explain the antimicrobial effect of

Next chapter 4 was dedicated to the pharmacological activity, however, it regards not only jellies but also other antimicrobial peptides, so I propose to mention about overview of the antimicrobial effect of royal jelly components. In subchapter 4.2. The authors described the wide antifungal effect of Jelleines I and II, but I couldn’t find information about III and IV. They have to be mentioned if they have this action or in literature is lack of information. A similar situation occurs in the case of the antifungal action of jelleines. Only in Table 3, we can find information that jelleines III and IV are inactive. This fact should be mentioned in the text also. Subchapters 4.3 and 4.4 as well as 4.5 ., and 4.6  are only information about jelleine 1.

Answer: Modification was performed as suggested by the reviewer.

Question #6: The chapter 5 regarding toxicity focused on jelleine -I. I couldn’t find information about the rest proteins.

Answer: As jelleine-I presents the most promising biological activity among all tested analogues, only it has been studied. None of the studies we sought mentioned the toxicity, stability or even pharmacological activity, other than antibacterial, of jelleines II-IV. This issue was highlighted in section “conclusions remakes”

Question #7: I would like to highlight that the conclusions are well prepared and indicate on potential role of jelleine-I as a promising prototype for new drug.

Answer: We thank the reviewer for the comment.

Question #8: The article lacks some formal information such as Author Contributions, Funding as well as Acknowledgments.

Answer: All this information is included in the version submitted to the periodic. However, editors remove them to ensure peer review is done blindly.

Reviewer 4 Report

Comments and Suggestions for Authors

The authors focused on Jelleine, a peptide derived from Royal jelly, and summarized its drug applications as Jelleine-I has been reported as a small molecule with the strongest antibacterial activity, low toxicity, and is relatively stable. This is a general review. There is a very detailed description of Jelleine-I, which is based on the biologically active substance derived from Royal jelly, and includes antibacterial, antifungal, antiinflammatory, and antitumor effects, and it shows how many activities this peptide has and its stability. I get the impression that it is easy to understand how good it is.

Author Response

General comments: The authors focused on Jelleine, a peptide derived from Royal jelly, and summarized its drug applications as Jelleine-I has been reported as a small molecule with the strongest antibacterial activity, low toxicity, and is relatively stable. This is a general review. There is a very detailed description of Jelleine-I, which is based on the biologically active substance derived from Royal jelly, and includes antibacterial, antifungal, antiinflammatory, and antitumor effects, and it shows how many activities this peptide has and its stability. I get the impression that it is easy to understand how good it is.

Answer: We thank the reviewer for the comments and we appreciate your willingness to review our manuscript.